# Sustainability of effects and secondary long-term outcomes: One-year follow-up of a cluster-randomized controlled trial to prevent maltreatment in institutional care

Tobias Hecker[1,2]*, Getrude Mkinga[1,2], Eva Hartmann[1], Mabula Nkuba[2,3], Katharin Hermenau[2,4]

1 Department of Psychology, Bielefeld University, Bielefeld, Germany, 2 vivo international, Konstanz, Germany, 3 Department of Educational Psychology and Curriculum Studies, Dar es Salaam University College of Education, Dar es Salaam, Tanzania, 4 Clinic of Child and Adolescent Psychiatry and Psychotherapy, Protestant Hospital Bethel, University Clinics OWL, Bielefeld University, Bielefeld, Germany

* tobias.hecker@uni-bielefeld.de

**Data Availability Statement:** All relevant data are available within the paper and its Supporting Information files (see S1 Data).

## Abstract

### Background

Many orphans in East Africa are living in institutional care facilities where they experience poor quality of care and ongoing maltreatment. We report on the extension of a cluster-randomized controlled trial aiming to replicate and show sustainability of previous found effects and to discover long-term effects of the intervention *Interaction Competencies with Children–for Caregivers* (ICC-C) 12-months after the intervention's conclusion.

### Methods

Conducting a robust 2x3 analysis of variance, we investigated the changes over time in the waitlist orphanages ($n = 75$, 62.7% female, $M_{age} = 37.63$ years, $SD_{age} = 11.81$), which participated in the intervention after first follow-up and in the initial intervention orphanages ($n = 81$, 61.7% female, $M_{age} = 38.73$ years, $SD_{age} = 11.94$).

### Results

The caregivers in the waitlist orphanages reported less reported levels of maltreatment ($d = -0.09$), fewer positive attitudes towards violent discipline ($d = -0.44$) and increased child-care knowledge ($d = 1.26$) three months after intervention, replicating our findings of the initial intervention condition. In addition, these effects were maintained in the intervention orphanages 12 months post intervention. Furthermore, we found long-term improvements in negative caregiver-child relationship ($d = -0.83$), caregivers' stress level ($d = -0.98$) and their mental health problems ($d = -0.61$).

**Funding:** TH was supported by The Bielefeld Young Researchers' Fund of Bielefeld University and vivo international. The funders had no role in study design, data collection and analysis, decision to publish, or preparation of the manuscript.

**Competing interests:** Tobias Hecker, Getrude Mkinga, and Katharin Hermenau were involved in the development of the intervention Interaction Competencies with Children – for Caregivers (ICC-C) tested in this study. There are no patents, products in development or marketed products associated with this research to declare. This does not alter our adherence to PLOS policies on sharing data and materials.

## Conclusions

The replication and maintenance of the intervention effects and first hints to additional long-term effects substantiates the effectiveness of ICC-C. As long as alternative care cannot be provided for all children in need, brief caregiver trainings can make an important contribution to enlarge the opportunities for many children.

## Trial registration

ClinicalTrials.gov, NCT03594617. Registered on 20 July 2018.

## Introduction

In 2015, the number of orphans globally was estimated to be 140 million [1], the majority of which living in Sub-Saharan Africa [2]. Besides loss due to AIDS/HIV or war, other parental illnesses, poverty, disabilities, and abandonment result in children growing up without their parents [3, 4]. Though family-based care generally promotes the best outcomes for children [5], current challenges in developing countries, such as the lack of financial resources, governmental support, and professional social-work infrastructure as well as an increasing number of children in need [4], result in the lack of family-based care settings [6] and the persistence of institutional care facilities short and medium term. In addition to advocacy for and introduction of family-based care, there is also a need for approaches which emphasize the improvement of institutional care [6].

Many institutional care settings in low- and middle-income countries are characterized by a lack of guidelines and quality control [7]. The caregivers are rarely specialized [8] or trained at all [3], and are stressed out by an unsustainable work load due to their poor working conditions and the structural circumstances at the institution [9, 10]. Due to the structural conditions and the lack of a warm, sensitive, enriched, and child-orientated environment, the disadvantages of children living in institutional care are self-evident: A great number of institutionalized orphaned children are characterized as being in fair or poor health [11], generally developed below average [12], at higher risk for cognitive delay [13] as well as insecure and disorganized attached [14]. However, it is not only the absence of adequate care, in the form of emotional or physical neglect, that can lead to impairments in the children's physical, mental and social development, also experiences of violence were associated with emotional and behavioral problems [15], a lower psychological quality of life [16], and epigenomic changes regarding stress regulation [17].

Violent forms of discipline are frequently used worldwide. In many Sub-Saharan African societies, violent discipline is considered a necessary aspect of a child's upbringing and is widely accepted [10, 18]. Positive correlations between the use of maltreatment and positive attitudes towards violence [19] as well as stress [20] were found among caregivers. When considering the limited training and working conditions of caregivers [3, 9, 10], it is not surprising that the rates of violence in institutional care are high [18]. Indeed, as long as alternative care cannot be provided for all children in need, childcare trainings may help to educate and to empower caregivers and, thus, provide a nonviolent and caring environment for children at risk, regardless of their caregiving situation [6].

Many studies on interventions implemented in childcare institutions have evaluated programs that provided structural changes in the institution or a social-emotional caregiver

training [9, 21, 22]. Trainings like these also showed additional positive effects on caregivers' stress and mental health: After receiving an intervention, the caregivers reported reduced job stress, anxiety, and depression [9]. The effects were stronger nine months after the intervention compared to four months after the intervention. Despite the existence of several caregiver trainings, there is a need for program enhancements as violence and abuse prevention are often not considered [7, 13]. To address this shortcoming, the intervention *Interaction Competencies with Children–for Caregivers (ICC-C)* was developed [10]. The feasibility and effectiveness of *ICC-C* has already been tested in the context of a two-arm cluster randomized controlled trial in Tanzanian orphanages with positive initial findings [23]. The caregivers who participated in the intervention reported a decreased use of maltreatment, less positive attitudes towards violent discipline, and improved childcare knowledge after three months as compared to the caregivers working in the waitlist orphanages. However, these promising findings need to be replicated and the sustainability of the effects remains unclear. Moreover, a three-months period might be too short to impact changes in variables like attachment, reduction of stress, and mental health problems because we expect that they would require an integration and stabilization of the training contents in daily work. Therefore, we extended our trial to a second follow-up. In the second follow-up the waitlist orphanages had also received the training, and 12 months had passed for the caregivers in the initial intervention orphanages who had received the intervention.

Our primary aims were to replicate and further investigate the promising findings in the caregivers' sample [23]. We expected ($a^1$) a significant decrease in caregiver reported maltreatment and ($b^1$) positive attitudes towards violence as well as ($c^1$) a significant increase in childcare knowledge in the waitlist orphanages from first to second follow-up. In the intervention orphanages, we expected ($a^2$) a significant decrease in caregiver-reported maltreatment and ($b^2$) attitudes towards violence as well as ($c^2$) a significant increase in childcare knowledge from first to second follow-up.

Our secondary aim was to test for the presence of delayed adjustments. To be specific, we expected (d) a significant decrease in negative caregiver-child relationship, (e) stress level and (f) mental health problems one year after training participation (from baseline to second follow-up) in the initial intervention orphanages.

## Methods

### Study design and setting

This parallel group cluster-randomized controlled trial involving all orphanages in Dar es Salaam city, Tanzania, was a continuation of the work by Hecker et al. [23]. In total, 24 orphanages were randomly assigned to the intervention or to the waitlist condition. A true random number service, http://www.random.org, was used for randomization and allocation purposes. The former waitlist attended the *ICC-C* intervention as well and at second follow-up, the treatment of the groups only differed in the time point of participation. Namely, the intervention orphanages (*n* = 81) participated in the training after baseline while the waitlist orphanages (*n* = 75) got access to the training after first follow-up (6 months after the intervention group.

### Sampling

Before baseline, we contacted all registered orphanages in the Dar es Salaam region, Tanzania, by informing the districts' welfare offices of the opportunity for the institutions to participate. Through the registered orphanages, we initiated contact with further unregistered orphanages to make sure to include every orphanage in Dar es Salaam region into our study. We recruited

the participants through lists of names provided by the orphanages. As the staff turnover was high, we decided to include new participants after first follow-up to portray the real conditions in the orphanages. Inclusion criteria for the caregivers required that the caregivers be of legal age (18 years) and that they had signed the written informed consent. The flow of participants is illustrated in Fig 1.

## Participants

**Orphanages.** Overall, the orphanages were very heterogeneous in terms of material and human resources, sponsors (public, religious, community-based), professionalism of management, number of caregivers, and children. However, it reflects the reality of life for orphans in Dar es Salaam. Out of the 24 institutions five orphanages were fully registered, 10 were in the process of registration and 9 orphanages were not registered. Between 13–121 children and adolescents ($Mdn$ = 46), ranging from 0 to 28 years of age (note that some young adults are supported for studies at university) were living in the orphanages.

**Caregivers.** The 156 participating caregivers (62% female) had a mean age of 38.20 years, $SD$ = 11.85, range: 19–66. Less than half (44% of $n$ = 130) had received any specialized training in the work with children, with an average training duration of 5.89 weeks ($SD$ = 12.48, range: 0–60, $n$ = 113) among those who had received such training. The participants had worked as a caregiver for an average of 7.04 years ($SD$ = 6.73, $n$ = 131), with a full range from one month to 31 years. In total, 36% of 152 caregivers reported living at the orphanage. The other 96 caregivers reported an average worktime of 67.03 hours per week ($SD$ = 3.57, range: 8–144). The caregivers were primarily responsible for $M$ = 9.06 children ($SD$ = 10.28, range: 1.79–17.50 (between institutions), $n$ = 111). See Table 1 for additional information about the participants.

## Procedure

**Assessment procedure.** Before the first assessment, the heads and caregivers of the institutions personally received explanations about the procedure and aims of the study. We collected the baseline data from August to October 2018. First follow-up took place from March to April 2019. The second follow-up was from January to March 2020. At all three time points, we conducted face-to-face interviews in Swahili which took place at the orphanages with an average duration of 30 minutes. In total, ten research assistants supported the data assessment. They were trained extensively before each assessment period. At all three time points, they were blinded to the caregivers' programmatic allocation.

**Intervention procedure.** After baseline, the intervention group participated in the training. In three training rounds, each with 31–43 caregivers, three trainers with a psychological background conducted the training, which was provided completely in Swahili. The training was free of charge and participants received free meals, drinks, and reimbursement for travel expenses in the amount of 5,000 TSH (approx. 2.17 USD) per day. After first follow-up, the waitlist group participated in the intervention as well. The procedure was identical to the training of the intervention group.

**Ethical considerations.** The study was performed in accordance with the ethical standards as laid down in the 1964 Declaration of Helsinki and its later amendments. We obtained ethical approval from the ethics review board of the University of Konstanz, Germany, and a research permit for Tanzania by the University of Dar es Salaam on behalf of the Tanzania Commission for Science and Technology. Informed consent was obtained from all individual participants included in the study.

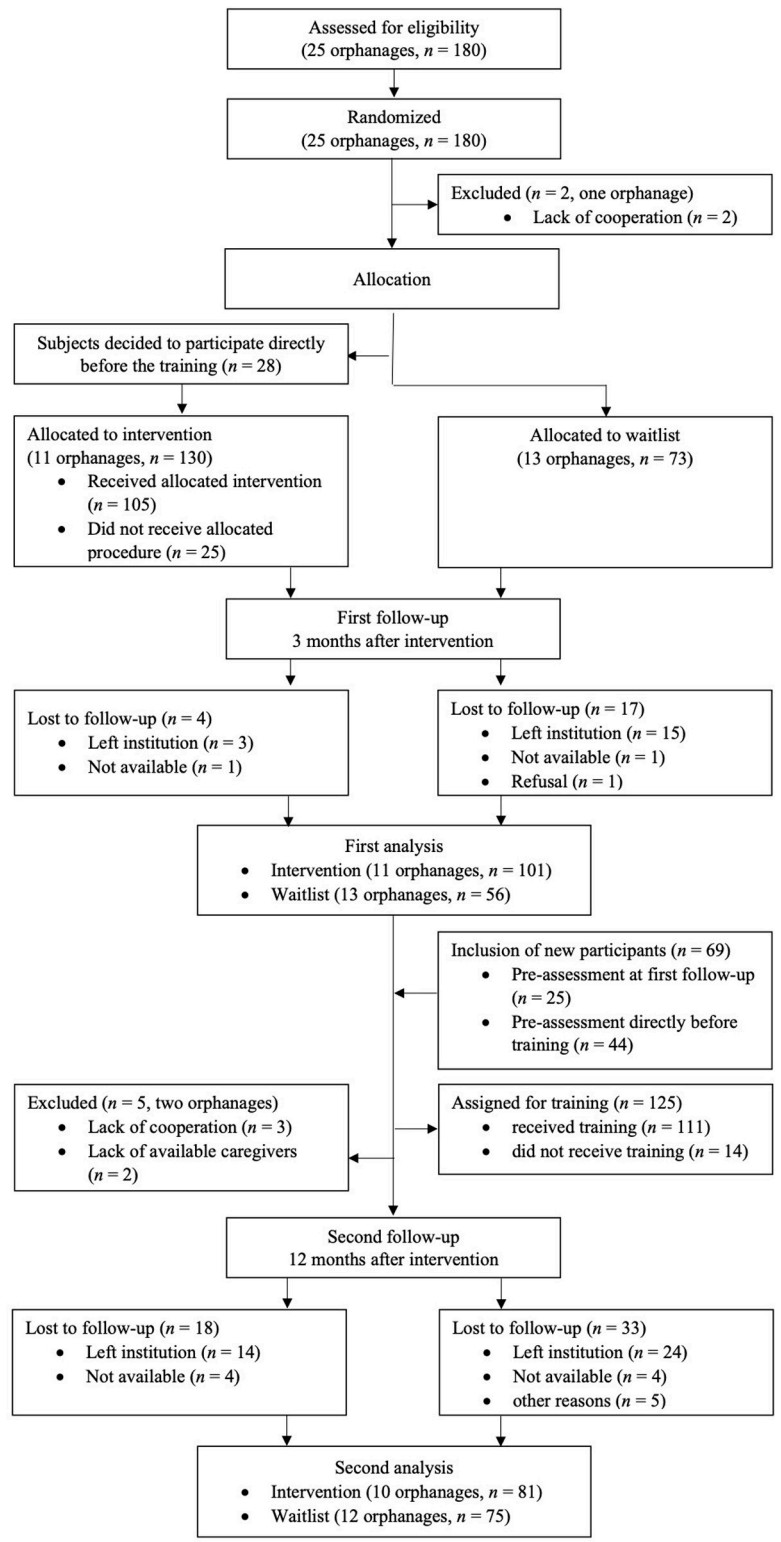

**Fig 1. Participant flow chart.**

**Table 1. Descriptive statistics of demographics of caregivers separated by intervention and waitlist orphanages.**

| | Intervention | | Waitlist | |
|---|---|---|---|---|
| | *N* | *n (%)* | *N* | *n (%)* |
| Gender (male) | 81 | 31 (38.3) | 75 | 28 (37.3) |
| Own children (yes) | 81 | 64 (79.0) | 75 | 45 (60.0) |
| Specialized training in childcare (yes) | 81 | 31 (38.3) | 49 | 27 (55.1) |
| Work hours [a] | | | | |
| Living at the orphanage | 79 | 33 (41.8) | 75 | 22 (30.1) |
| Fulltime (> 40h per week) | 79 | 30 (38.0) | 75 | 37 (50.7) |
| Part time | 79 | 6 (7.6) | 75 | 8 (11.0) |
| Voluntary work | 79 | 10 (12.7) | 75 | 6 (8.2) |
| Other sources of income (yes) | 81 | 37 (45.7) | 74 | 30 (41.3) |
| Monthly income 100 USD or higher | 78 | 17 (21.8) | 71 | 26 (36.6) |
| Formal contract (yes) | 79 | 24 (30.4) | 72 | 27 (37.5) |
| | *N* | *M (SD)* | *N* | *M (SD)* |
| Age | 81 | 38.73 (11.94) | 75 | 37.63 (11.81) |
| Years of education | 81 | 10.00 (3.26) | 74 | 10.70 (3.97) |
| Years worked as a caregiver [b] | 81 | 6.87 (5.62) | 50 | 7.33 (8.27) |
| Duration of childcare-related education, in weeks [b] | 70 | 3.73 (9.42) | 43 | 9.40 (15.78) |
| Interaction time with children (hours per day) | 67 | 1.64 (1.11) | 67 | 1.87 (1.61) |
| Time for housekeeping activities (hours per day) | 66 | 2.35 (2.63) | 59 | 2.81 (2.68) |
| Household income/month (in USD) | 78 | 79.07 (112.91) | 71 | 93.13 (88.89) |
| Days of holiday per year | 63 | 18.81 (22.10) | 69 | 19.29 (14.82) |
| Work satisfaction [c] | 81 | 2.49 (0.81) | 73 | 2.40 (0.94) |
| Self-rating child-care knowledge [c] | 80 | 1.95 (0.87) | 75 | 1.73 (0.98) |

*Note. N* = number of participants who answered the item, *n* = absolute number of participants who answered with yes, *M* = mean, *SD* = Standard Deviation. Because of cluster effects no inference statistical analysis is made.

[a] Item 'Work hours': As an official concept of weekly hours of work does not exist in Tanzania, this item is based on personal ratings, additional notes on the questionnaires, and oral statements within the assessment situation.

[b] Items 'Years worked as a caregiver' and 'Duration of childcare-related education, in weeks': only measured at baseline; the new participants that joined at first follow-up did not answer these items.

[c] Items 'work satisfaction' and 'self-rating child-care knowledge': ratings on a 4-point Likert Scale with a possible range from 0–3.

## Intervention

The two-week (2x 5.5 days) training workshop *Interaction Competencies with Children–for Caregivers* is based on attachment, behavioral, and social learning theories as well as on the parenting guidelines of the American Academy of Pediatrics [24]. ICC-C has been inspired by the FairstartGlobal training concept [25], and the maltreatment prevention components were grounded in the work of Dreikurs [26]. Through the medium of warm, sensitive and reliable caregiver-child relationships and nonviolent, warm and sensitive caregiving strategies the training aims at improving the quality of care and preventing maltreatment [3]. *ICC-C* consists of seven core components: *child development*, *caregiver-child relationship*, *effective caregiving strategies*, *maltreatment prevention*, *supporting burdened children*, *child-centered institutional care* and *teamwork and supervision*. Further details can be found in Hecker et al. [3].

## Materials

**Reported levels of maltreatment.** We used the Conflict Tactics Scale Parent-Child version (CTSCP) to assess the use of maltreatment by caregivers [27]. We adapted the instructions for caregivers in orphanages and added three items to the *neglect* subscale (*How often in the past month have you . . . had no time to play together with the children? . . . had no interest in listening and talking to a child? . . . had no time to ask a child whether he/she has a problem?*). The caregivers stated their answers on a 7-point Likert scale with *0 = never* to *6 = more than 20 times per month*. We recoded the values, so that each item represented the mid number of a specific violent method per month based on Straus et al. [27]. The sum score of the scales for *emotional violence* (five items), *physical violence* (13 items) and *neglect* (five items) acted as our outcome measure. Cronbach's alpha for the sum score at baseline indicated an acceptable reliability ($\alpha = .74$).

**Attitudes towards violence.** We measured attitudes towards violent discipline methods using the CTSCP with an adapted instruction which was used in previous studies in East-Africa [28]. The caregiver stated their answers on a 4-point Likert scale with *0 = never OK* and *3 = always OK*. The sum score of the scales for *emotional* and *physical violence* acted as our outcome measure. Internal consistency in this study was acceptable with $\alpha = .70$ for the sum score at baseline.

**Childcare knowledge.** To investigate the extent of childcare knowledge, we developed a multiple-choice questionnaire with 11 items which tested the caregivers on the content of the training (see S1 Table for details). Each item existed of three to four right or wrong answers. Our outcome measure was the number of correct answers (range: 0–40).

**Caregiver-child relationship.** We measured the caregiver-child relationship using an adapted version of the 15-item *People in my Life* (PIML) questionnaire [29]. The caregivers reported their answers on a 3-point Likert scale with *0 = not true* and *2 = certainly true*. Our outcome measure was the sum score in which higher values indicated a more negative caregiver-child relationship (range: 0–30). Internal consistency in this study was $\alpha = .63$ for the sum score at baseline.

**Stress level.** Using the Copenhagen Burnout Inventory (CBI) [30], we measured *personal* (six items), *work-related* (seven items), and *child-related burnout* (six items) on a 5-point Likert scale (*0 = never* and *4 = always*). Our outcome measure was the average score of all items (range: 0–100). The reliability in our study ($\alpha = .91$ for the baseline value) was satisfactory.

**Mental problems.** The Brief Symptom Inventory (BSI-18) consists of the three six-item scales *somatization*, *depression*, and *anxiety* [31]. Items were rated on a 5-point Likert scale with *0 = not at all* and *4 = extremely*. The sum score of all items acted as our outcome measure. Internal consistency in this study was good with $\alpha = .89$ for the sum score at baseline.

## Data analysis

Although the orphanages showed a high variation regarding the outcome measures at baseline, we decided not to use multilevel modelling due to the decreased number of cases at second follow-up. To consider the orphanages' heterogeneity, we zeroed the orphanage-specific baseline values for each outcome variable. Using this value transformation or standardization of values, all orphanages started at the same level. This made it easier to visualize the differences, because the changes within orphanages could be compared across orphanages. The values for first and second follow-up showed the respective difference to the baseline value. To analyze the data, we conducted a two-way mixed analysis of variance (2x3 ANOVA) for each outcome variable with the independent variables *Allocation* (intervention versus waitlist) and *Time* (baseline versus first follow-up versus second follow-up). We chose a robust ANOVA version using 20%

trimmed means [32, 33] since the distributional assumptions were not met and we had to deal with various outliers. After running the ANOVA, we calculated robust two-sided pairwise comparisons separately for the intervention and the waitlist orphanages to get a deeper insight in the group-specific changes over time [32, 34]. Partial eta squared served as an effect size for the omnibus tests, in which .01, .06 and .14 mark the thresholds for small, medium and large effects [35, 36]. For the multiple comparisons we chose Cohen's *d* for dependent samples, in which 0.14, 0.35 and 0.57 indicate a small, medium or large effect [35, 37].

For data preparation and descriptive results, we used IBM SPSS Statistics Version 21 (IBM Corp., Armonk, N.Y., USA). For inferential analysis, we used RStudio Version 3.6.1. An a priori power analysis ($\alpha = .05$, power = .80, $f = 0.25$) with G*Power software revealed a required sample size of at least $N = 54$ for the interaction effect in case of zero correlation among repeated measures. Following the expectation that the new participants did not undergo changes in the outcome variables before joining the study, their baseline values were replaced with their values at first follow-up (*next observation carried backward*). Only caregivers with data from all three time points were included in the analysis.

## Results

S2 Table shows the descriptive statistics for the main outcomes. We used the trimmed and transformed means for calculating the inferential statistics. The results of the robust ANOVA are presented in Table 2. We found significant training effects (*Allocation* X *Time* interaction) in reported levels of maltreatment, attitudes towards violence, and childcare knowledge. Subsequently, we conducted robust pairwise comparisons to answer our primary hypotheses (see also Table 3 and Fig 2). Even though the training effect did not reach significance in caregiver-child relationship, stress level, or mental health problems, we decided to calculate pairwise comparisons because the effect sizes indicated at least a small practical significance (see also Table 4 and Fig 3).

**Table 2. Robust ANOVA summary table.**

| | Reported levels of maltreatment | | | | Attitudes towards violence | | | |
|---|---|---|---|---|---|---|---|---|
| Source | *df* | *F* | *p* | $\eta_p^2$ | *df* | *F* | *p* | $\eta_p^2$ |
| Allocation | (1, 94.30) | 5.52 | .021* | .06 | (1, 91.66) | 19.91 | < .001*** | .18 |
| Time | (2, 68.82) | 24.14 | < .001*** | .41 | (2, 79.47) | 43.86 | < .001*** | .52 |
| Allocation X Time | (2, 68.82) | 6.37 | .003** | .17 | (2, 79.47) | 30.99 | < .001*** | .44 |
| | Childcare knowledge | | | | Caregiver-child relationship | | | |
| Source | *df* | *F* | *p* | $\eta_p^2$ | *df* | *F* | *p* | $\eta_p^2$ |
| Allocation | (1, 77.42) | 10.03 | .002** | .11 | (1, 88.55) | 2.65 | .107 | .03 |
| Time | (2, 73.59) | 40.21 | < .001*** | .52 | (2, 77.08) | 10.25 | < .001*** | .21 |
| Allocation X Time | (2, 73.59) | 27.16 | < .001*** | .42 | (2, 77.08) | 1.44 | .244 | .04 |
| | Stress level | | | | Mental problems | | | |
| Source | *df* | *F* | *p* | $\eta_p^2$ | *df* | *F* | *p* | $\eta_p^2$ |
| Allocation | (1, 92.17) | 4.04 | .047* | .04 | (1, 95.13) | 0.18 | .675 | < .01 |
| Time | (2, 74.35) | 14.54 | < .001*** | .28 | (2, 81.56) | 6.60 | .002** | .14 |
| Allocation X Time | (2, 74.35) | 2.02 | .141 | .05 | (2, 81.56) | 0.54 | .587 | .01 |

*Note.* * $p < .05$,

** $p < .01$,

*** $p < .001$. $\eta_p^2$ = partial eta squared.

**Table 3. Pairwise comparisons of the training effect in the primary outcomes separated by intervention and waitlist orphanages.**

| Comparisons | Intervention | | | | Waitlist | | | |
|---|---|---|---|---|---|---|---|---|
| | $\widehat{\psi}$ | 95% CI | $p_{emp}$ ($p_{crit}$) | $d$ | $\widehat{\psi}$ | 95% CI | $p_{emp}$ ($p_{crit}$) | $d$ |
| **Reported levels of maltreatment** | | | | | | | | |
| BA–FU 1 | −17.14* | [−23.64, −10.65] | < .001 (.025) | −1.40 | −5.80• | [−11.94, 0.34] | .023 (.025) | −0.40 |
| BA–FU 2 | −16.73* | [−22.28, −11.19] | < .001 (.017) | −1.76 | −11.00* | [−19.55, −2.45] | .003(.017) | −0.61 |
| FU 1 –FU 2 | 0.06 | [−1.47, 1.59] | .921 (.050) | 0.02 | −1.07 | [−5.86, 3.73] | .582 (.050) | −0.09 |
| **Attitudes towards violence** | | | | | | | | |
| BA–FU 1 | −2.94* | [−3.99, −1.89] | < .001 (.017) | −1.38 | −0.33 | [−0.76, 0.10] | .061 (.050) | −0.27 |
| BA–FU 2 | −2.68* | [−3.71, −1.63] | < .001 (.025) | −1.48 | −1.64* | [−2.62, −0.68] | < .001 (.017) | −0.82 |
| FU 1 –FU 2 | 0.06 | [−0.44, 0.57] | .765 (.050) | 0.06 | −0.76• | [−1.52, 0.01] | .018 (.025) | −0.44 |
| **Childcare knowledge** | | | | | | | | |
| BA–FU 1 | 3.16* | [2.07, 4.26] | < .001 (.017) | 1.32 | 0.04 | [−0.46, 0.55] | .826 (.050) | 0.04 |
| BA–FU 2 | 3.18* | [1.78, 4.59] | < .001 (.025) | 1.24 | 2.98* | [1.91, 4.04] | < .001 (.025) | 1.28 |
| FU 1 –FU 2 | 0.02 | [−0.90, 0.94] | .956 (.050) | 0.01 | 2.78* | [1.81, 3.75] | < .001 (.017) | 1.26 |

*Note.* $\widehat{\psi}$ = test statistic. Hochberg's approach was used to control for the family-wise error. Comparisons reached significance (*) if the 95% confidence interval (95% CI) did not include zero and the empirical *p*-value ($p_{emp}$) did not exceed the critical *p*-value ($p_{crit}$). Fulfilling only one criterion led to marginal significance (•). *d* = Cohen's *d* for dependent samples. BA = baseline, FU 1 = first follow-up, FU 2 = second follow-up.

## Primary outcomes

**Reported levels maltreatment.** There was neither a significant difference between first and second follow-up in the waitlist orphanages (a[1]), $\widehat{\psi}$ = −1.07 (−5.86, 3.73), *p* >.05, *d* = −0.09, or in the intervention orphanages (a[2]), $\widehat{\psi}$ = 0.06 (−1.47, 1.59), *p* >.05, *d* = 0.02.

**Attitudes towards violence.** In the waitlist orphanages (b[1]), the difference between first and second follow-up reached marginal significance, $\widehat{\psi}$ = −0.76 (−1.52, 0.01), *p* < .025. The effect size was medium with *d* = −0.44. We did not find a significant difference between first and second follow-up in the intervention orphanages (b[2]), $\widehat{\psi}$ = 0.06 (−0.44, 0.57), *p* >.05, *d* = 0.06.

**Childcare knowledge.** In the waitlist orphanages (c[1]), we found a significant difference between first and second follow-up, $\widehat{\psi}$ = 2.78 (1.81, 3.75), *p* < .017 with a large effect size (*d* = 1.26). There was no significant difference between first and second follow-up in the intervention orphanages (c[2]), $\widehat{\psi}$ = 0.02 (−0.90, 0.94), *p* >.05, *d* = 0.01.

## Secondary outcomes

**Caregiver-child relationship.** In the intervention orphanages (d), there was a significant difference between baseline and second follow-up, $\widehat{\psi}$ = 1.49 (−2.39, −0.59), *p* < .017. The effect size was large with *d* = −0.83.

**Stress level.** In the intervention orphanages (e), the comparison between baseline and second follow-up reached significance, $\widehat{\psi}$ = −8.25 (−12.07, −4.44), *p* < .017 with a large effect size of *d* = −0.98.

**Mental health problems.** In the intervention orphanages (f), there was a significant difference between baseline and second follow-up, $\widehat{\psi}$ = −2.53 (−4.23, −0.83), p < .025. The effect size was large with d = −0.61.

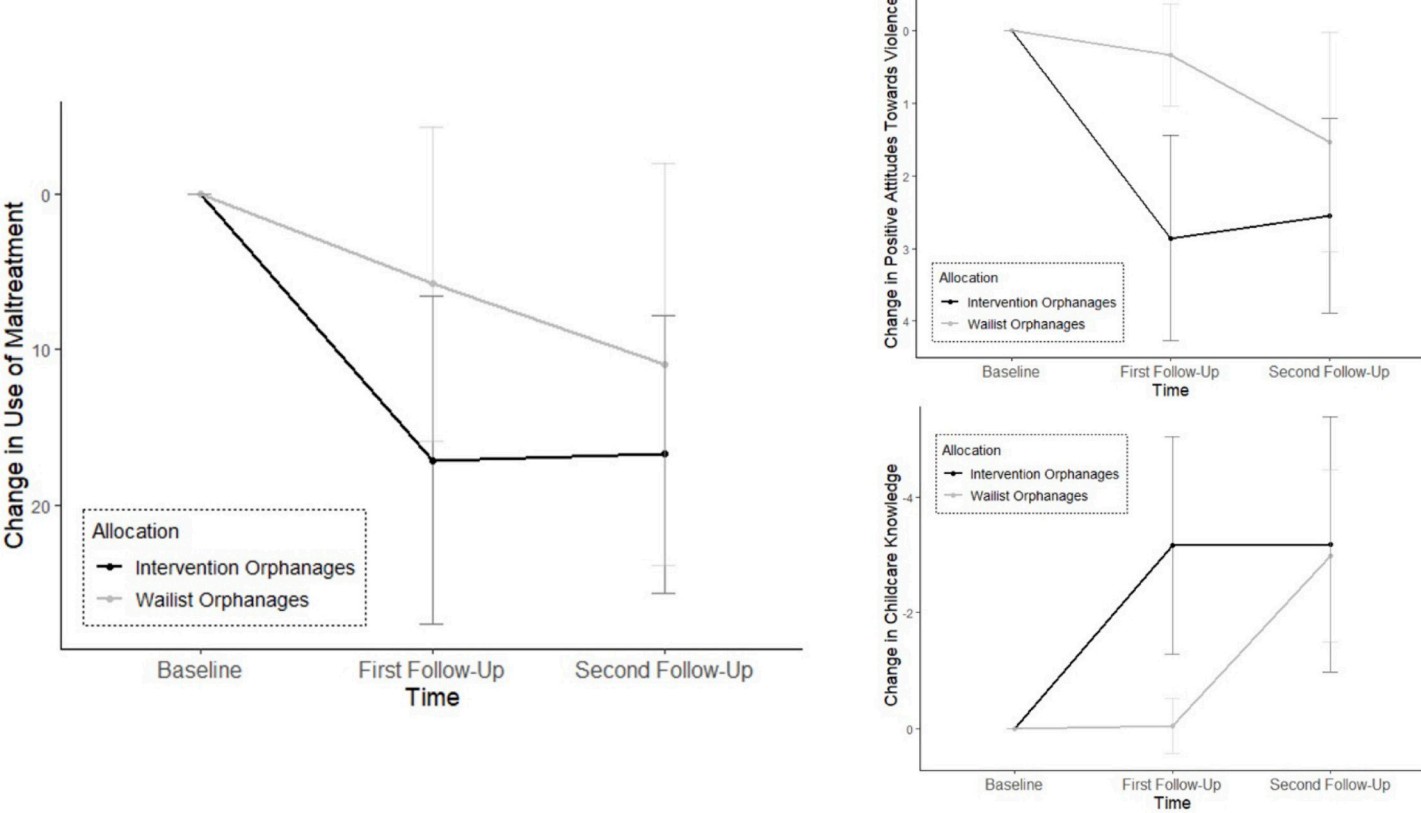

**Fig 2. Change in primary outcomes over time separated by intervention and waitlist orphanages.** The graphic displays the trimmed and transformed means ($M_{ab}$) and standard deviations ($SD_{ab}$) for use of maltreatment, positive attitudes towards violence and childcare knowledge at all three time points. The values for first and second follow-up represent the respective difference to the baseline value. A positive difference indicates a decrease while a negative difference stands for an increase of the respective outcome measure over time. Note that the y-axis was mirrored for interpretative ease. Across all primary outcome measures, a clear picture emerges while the values of the intervention orphanages remain stable between first and second follow-up, the waitlist orphanages show decreases in use of maltreatment and positive attitudes towards violence as well as an increase in childcare knowledge after their participation in *ICC-C*.

## Discussion

The replication of the findings of Hecker et al. [23] was successful: After participating in the training, the caregivers of the waitlist orphanages showed similar changes, which were previously observed in the initial intervention orphanages. Namely, they showed (a[1]) a decrease in caregiver-reported levels of maltreatment with a small effect ($d = -0.09$), which, however, has not reached statistical significance, (b[1]) a significant decrease in positive attitudes towards violence with a medium effect ($d = -0.44$) and (c[1]) a significant increase in childcare knowledge with a large effect ($d = 1.26$).

The present study also demonstrated the 12-month sustainability of the training effects in the intervention orphanages. Even though we did not reveal further significant improvements in our outcome measures, the previously identified changes remained stable. Namely, the caregivers of the intervention orphanages maintained reduced reported levels of maltreatment (a[2]), reduced positive attitudes towards violence (b[2]), and heightened childcare knowledge (c[2]).

We partly succeeded in producing new findings concerning variables which might show changes only after a longer time period. Despite the lack of significant interaction effects in the ANOVA, the effect sizes for negative caregiver-child relationship ($\eta_p^2 = .04$), stress level ($\eta_p^2 = $

**Table 4. Pairwise comparisons of the training effect in the secondary outcomes separated by intervention and waitlist orphanages.**

| Comparisons | Intervention $\widehat{\psi}$ | 95% CI | $p_{emp}$ ($p_{crit}$) | $d$ | Waitlist $\widehat{\psi}$ | 95% CI | $p_{emp}$ ($p_{crit}$) | $d$ |
|---|---|---|---|---|---|---|---|---|
| **Caregiver-child relationship** | | | | | | | | |
| BA–FU 1 | −0.90 | [−1.93, 0.14] | .036 (.025) | −0.41 | −0.16 | [−0.66, 0.35] | .445 (.025) | −0.15 |
| BA–FU 2 | −1.49* | [−2.39, −0.59] | < .001 (.017) | −0.83 | −0.89 | [−1.84, 0.06] | .025 (.017) | −0.44 |
| FU 1 –FU 2 | −0.82 | [−1.85, 0.22] | .057 (.050) | −0.41 | −0.27 | [−1.37, 0.84] | .550 (.050) | −0.12 |
| **Stress level** | | | | | | | | |
| BA–FU 1 | −4.33• | [−8.73, 0.08] | .019 (.025) | −0.52 | −1.40 | [−3.33, 0.53] | .078 (.025) | −0.23 |
| BA–FU 2 | −8.25* | [−12.07, −4.44] | < .001 (.017) | −0.98 | −4.06 | [−8.34, 0.23] | .023 (.017) | −0.43 |
| FU 1 –FU 2 | −3.26 | [−8.01, 1.50] | .096 (.050) | −0.30 | −0.52 | [−4.41, 3.36] | .740 (.050) | −0.05 |
| **Mental problems** | | | | | | | | |
| BA–FU 1 | −0.47 | [−1.82, 0.88] | .394 (.050) | −0.10 | −0.62 | [−1.62, 0.37] | .127 (.025) | −0.22 |
| BA–FU 2 | −2.53* | [−4.23, −0.83] | < .001 (.025) | −0.61 | −1.69 | [−4.03, 0.65] | .080 (.017) | −0.38 |
| FU 1 –FU 2 | −1.59* | [−2.53, −0.65] | < .001 (.017) | −0.54 | −0.09 | [−1.42, 1.24] | .869 (.050) | −0.02 |

*Note.* $\widehat{\psi}$ = test statistic. Hochberg's approach was used to control for the family-wise error. Comparisons reached significance (*) if the 95% confidence interval (95% CI) did not include zero and the empirical *p*-value ($p_{emp}$) did not exceed the critical *p*-value ($p_{crit}$). Fulfilling only one criterion led to marginal significance (•). The pairwise comparisons for caregiver-child relationship, stress level, and mental health problems were calculated in an exploratory fashion due to the lack of a significant interaction effect. $d$ = Cohen's $d$ for dependent samples. BA = baseline, FU 1 = first follow-up, FU 2 = second follow-up.

.05), and mental problems ($\eta_p^2$ = .01) indicated a practical relevance. In pairwise comparisons, we revealed significant decreases in (d) negative caregiver-child relationship ($d = −0.83$), (e) stress level ($d = −0.98$), and (f) mental health problems ($d = −0.61$) for the intervention orphanages one year after training participation.

The fact that we replicated the previously known training effects on caregiver-reported levels of maltreatment, positive attitudes towards violence, and childcare knowledge in the waitlist orphanages supports the conclusion that the *ICC-C* intervention reduces maltreatment and improves care quality. Overall, our results align with other empirical studies indicating that caregiver trainings can be fruitful even under challenging circumstances [9, 10, 21–23, 28]. Considering the high prevalence of violence in Tanzania [38, 39] and the positive correlation of affirmative attitudes towards violent discipline and the use of physical maltreatment [19, 20], our results are highly relevant for preventing children from experiencing further abuse by their caregivers. Beyond that, our findings suggest that *ICC-C* can be implemented in orphanages with heterogenous conditions. Even in institutions with limited resources, our ICC-C could be implemented successfully. Based on the present findings, *ICC-C* shows potential to be expanded to a broad target group.

However, the findings of the present study also indicate potential limitations. The waitlist orphanages did not show as large a decrease in caregiver-reported levels of maltreatment and positive attitudes towards violence as we had expected. This could be attributed to the slight changes in these outcome measures which we observed in the waitlist orphanages even before they had participated in the training (Fig 2). Premature change in control groups during a waiting-only condition can be explained by the reactivity of measurement or the participants' motivation, self-reflection, and behavior modification simply because of their participation in a study [40]. The waitlist caregivers' self-monitoring concerning violent discipline measures prior to the training might be a reason why *ICC-C* did not have the same impact on their behavior and attitudes compared to the initial intervention orphanages. Unrealistic expectations about self-change can lead to frustration and discouragement [41]. It is possible that false

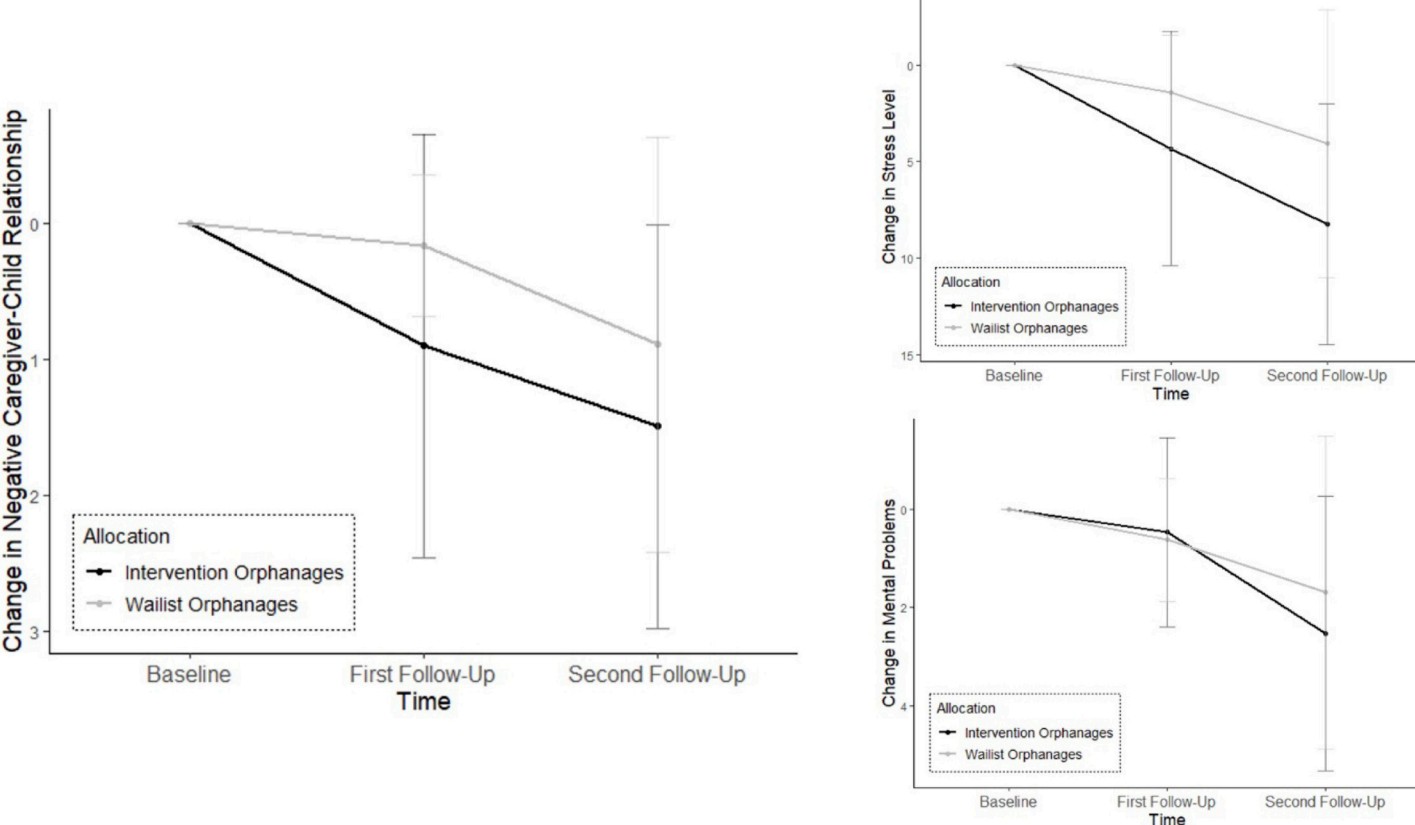

**Fig 3. Change in secondary outcomes over time separated by intervention and waitlist orphanages.** The graphic displays the trimmed and transformed means ($M_{ab}$) and standard deviations ($SD_{ab}$) for negative caregiver-child relationship, stress level and mental problems at all three time points. The values for first and second follow-up show the respective difference to the baseline value. A positive difference indicates a decrease while a negative difference stands for an increase of the respective outcome measure over time. Note that the y-axis was mirrored for a more intuitive understanding. The graphic illustrates: One year after participating in *ICC-C*, the intervention orphanages show significant decreases in negative caregiver-child relationship, stress level and mental problems.

hopes about the speed and feasibility of self-change hindered a larger decrease in the reported levels of maltreatment and positive attitudes towards violence. Another explanation could also be that participants in the waiting list group were also already familiar with the assessment tools in the follow-up interviews and were thus more likely to be concerned about positive self-presentation.

The increase in childcare knowledge in the waitlist orphanages after receiving *ICC-C* was as hypothesized. Since we measured this outcome using a test about the training contents, the change over time did not underlie the mentioned effects for reactive measurements [35] and was rarely influenced by other factors besides the training participation. Providing information can be a strong predictor for developing skills [42] and knowledge is an important pre-condition of behavior and a determinant for behavioral change [43]. The improved knowledge provides a base for heightening the quality of childcare in orphanages.

Our study was the first to show the sustainability of *ICC-C*. The maintenance of our observed training effects on caregiver-reported levels of maltreatment, positive attitudes towards violence, and childcare knowledge over a period of one year is in line with the results of the study of The St. Petersburg-USA Orphanage Research Team (2008) which also observed lasting improvements after a caregiver training. It must be pointed out that their intervention was much more extensive than *ICC-C* as it also covered structural changes within the childcare

institutions [9]. It is particularly noteworthy that changed attitudes persisted over time in our study as the presence of positive attitudes toward violent discipline has been identified as a robust predictor of violent behavior [44]. The fact that we were able to show enduring training effects using the two-week *ICC-C* training workshop alone indicates that even brief intervention approaches can initiate long-term changes. Nevertheless, it is of central importance to examine whether the self-reported behavioral changes are reflected in changed behavior. In the future, it will be important to include more objective observational measures and the reports of the affected children in the evaluation of ICC-C and comparable interventions.

Our aim to discover improvements in caregiver-child relationship, stress level, and mental problems over a 12-month period in the intervention orphanages was met to some extent. Despite the lack of significant interaction effects, we observed changes from baseline to second follow-up in these outcomes. Yet, they were not as large as could be expected from previous studies [9, 22]. As mentioned above, the St. Petersburg-USA Orphanage Research Team [9] implemented a caregiver training plus additional structural changes whose effects exceeded those of a training-only condition. Even though structural changes like familywise care settings are discussed during *ICC-C*, they may not necessarily transferred into the daily work without support on the management or governmental level [3]. Caregivers need not only a provision of knowledge and strategies, but also the time and resources to act in accordance with their skills and information [8]. Nevertheless, we found small effect sizes for the interaction effects showing evidence of a practical significance of our findings. Even though they were smaller than expected, our results of improvements are notable against the background of the current state of research: It is known that a good caregiver-child relationship can prevent children from experiencing several negative outcomes [11, 13, 14]. Additionally, reducing a caregiver's stress can lower their use of maltreatment [20] and improve their mental health [30]. Facing the fact that *stress* and *mental health* were not primarily addressed in the *ICC* approach, we consider these positive secondary effects as encouraging for further research.

Reaching a broad and heterogenous target group, our brief intervention promises to sustainably succeed in improving the quality of care and preventing maltreatment. As long as institutionalization still affects millions of children worldwide [4], *ICC-C* provides a chance to improve the opportunities for children in need without the availability of significant financial, organizational, or temporal resources. It may lay the foundation for a warm and secure child development regardless of the level of resources which children and caregivers in institutionalized care may have available.

Some limitations need to be acknowledged: Caregivers were not blinded, and social desirability cannot be ruled out as we assessed only self-report data. Objective assessments like observations are needed to address this problem. Moreover, the findings reported here relate exclusively to caregivers' self-reports. Additional reports from the affected children themselves would have been less biased and would be essential to include in future studies. The inclusion of new participants after first follow-up may be viewed critically because the sample from second follow-up might not represent the initial sample anymore. Notwithstanding, we succeeded in reproducing the changes from baseline to first follow-up [23], which indicated clear similarities between the two samples. With the inclusion of new participants, we captured the natural flow of caregivers within an orphanage, increasing the present study's external validity. The power of our multiple comparisons might be underestimated due to the calculation of more comparisons than needed for our hypotheses. Through adjusting the *p*-value and using two-tailed tests, we further decreased the probability of finding true results rejecting our null hypotheses. We chose pairwise comparisons instead of linear contrasts because we were interested in the results of the comparisons between all time points. We can conclude that even with our conservative approach, we found significant results which underline the practical

significance of our findings. The generalizability of our present results to institutions in other regions of the globe is limited since we included orphanages from only one region in Tanzania. Nevertheless, the heterogeneity across the orphanages in our studies indicates that our intervention is effective in a diverse target group. In addition, *ICC* has demonstrated its effectiveness in other regions and contexts in previous studies [10, 28]. Structural factors in individual orphanages, such as such as material and personnel equipment, type of sponsorship, etc., could have influenced effects. Therefore, it would be interesting to determine and consider the influence of possible structural factors in the future.

## Conclusion

Institutionalized children from low- and middle-income countries are faced with the risk of experiencing neglect and violence [11, 13–15]. To improve the living conditions in childcare institutions, burdened caregivers need to be supported and educated [8]. The preventative intervention *Interaction Competencies with Children–for Caregivers (ICC-C)* enriches existing interventions as it also addresses the prevention of maltreatment in addition to improving the quality of care [3]. In this study, we found further evidence for the effectiveness of *ICC-C* by replicating the three-month intervention effects [23] regarding a decrease in caregiver-reported levels of maltreatment and positive attitudes towards violence as well as an increase in childcare knowledge. Furthermore, our findings provide initial evidence for the sustainability of these effects over a 12-month period. Additionally, we found improvements in caregiver-child relationship, caregiver stress level, and caregiver mental health one year after training participation. Overall, we suggest further implementing and evaluating *ICC-C* to enhance the current situation for as many children and caregivers as possible. As long as alternative care settings cannot be provided for all institutionalized children, caregiver trainings like *ICC-C* contribute to make the most of the difficult circumstances for children in need. Nevertheless, we must note that training for caregivers can only be one piece of the puzzle that protects children in institutional care from violence and neglect in the short and medium term. The goal of child protection efforts should be the establishment of high-quality family-based care, which should eventually make institutional care unnecessary [5].

## Supporting information

**S1 Table. Assessment of childcare knowledge.**
(DOCX)

**S2 Table. Descriptive statistics of outcome variables.**
(DOCX)

**S1 Data. Data files.**
(SAV)

## Acknowledgments

We are very grateful to the sponsors and management of the institutions involved, as well as to all study participants for their participation in the study. We thank our research team, especially Simeon Mgode, for their great support during the study implementation.

## Author Contributions

**Conceptualization:** Tobias Hecker, Katharin Hermenau.

**Data curation:** Tobias Hecker, Getrude Mkinga, Mabula Nkuba.

**Formal analysis:** Tobias Hecker, Eva Hartmann.

**Funding acquisition:** Tobias Hecker.

**Investigation:** Tobias Hecker, Getrude Mkinga, Mabula Nkuba.

**Methodology:** Tobias Hecker, Eva Hartmann.

**Project administration:** Tobias Hecker, Getrude Mkinga, Mabula Nkuba.

**Resources:** Tobias Hecker.

**Software:** Tobias Hecker, Eva Hartmann.

**Supervision:** Tobias Hecker, Katharin Hermenau.

**Validation:** Katharin Hermenau.

**Visualization:** Eva Hartmann.

**Writing – original draft:** Tobias Hecker, Eva Hartmann.

**Writing – review & editing:** Tobias Hecker, Eva Hartmann, Mabula Nkuba, Katharin Hermenau.

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
