## [Decision Letter · Decision Letter 0]

11 Jan 2022

PGPH-D-21-00959

Sustainability of effects and secondary long-term outcomes: One-year follow-up of a cluster-randomized controlled trial to prevent maltreatment in institutional care

Dear Dr. Hecker,

Thank you for submitting your manuscript to PLOS Global Public Health. After careful consideration, we feel that it has merit but does not fully meet PLOS Global Public Health’s publication criteria as it currently stands. Therefore, we invite you to submit a revised version of the manuscript that addresses the points raised during the review process.

We look forward to receiving your revised manuscript.

Kind regards,

Martin Heine

Academic Editor

Journal Requirements:

1. Please update the completed 'Competing Interests' statement. If you have no competing interests to declare, please state: “The authors have declared that no competing interests exist.”

2. Please provide a complete Data Availability Statement in the submission form, ensuring you include all necessary access information or a reason for why you are unable to make your data freely accessible. Note that it is not acceptable for the authors to be the sole named individuals responsible for ensuring data access.

PLOS defines a study's minimal data set as the underlying data used to reach the conclusions drawn in the manuscript and any additional data required to replicate the reported study findings in their entirety. Any potentially identifying patient information must be fully anonymized. 

If your research concerns only data provided within your submission, please write “All data are in the manuscript and/or supporting information files” as your Data Availability Statement.

3. Please provide separate figure files in .tif or .eps format only and ensure that all files are under our size limit of 20MB.

4. Please include a legend for Figure 1 in your manuscript.

5. Thank you for providing an Ethics Statement for your study. Currently, your ethics statement appears outside of your Methods section of  your manuscript (introduction, results, conclusions, acknowledgements). Please ensure that the Ethics statement appears only in your Methods section and remove any other occurances.

6. We have noticed that you have uploaded supporting information but you have not included a list of legends.  Please add a full list of legends for all supporting information files (including figures, table and data files) after the references list.

Additional Editor Comments (if provided):

Reviewers' comments:

Reviewer's Responses to Questions

**Comments to the Author**

1. Does this manuscript meet PLOS Global Public Health’s publication criteria? Is the manuscript technically sound, and do the data support the conclusions? The manuscript must describe methodologically and ethically rigorous research with conclusions that are appropriately drawn based on the data presented.

Reviewer #1: Yes

Reviewer #2: Yes

2. Has the statistical analysis been performed appropriately and rigorously?

Reviewer #1: Yes

Reviewer #2: I don't know

3. Have the authors made all data underlying the findings in their manuscript fully available (please refer to the Data Availability Statement at the start of the manuscript PDF file)?

Reviewer #1: Yes

Reviewer #2: No

4. Is the manuscript presented in an intelligible fashion and written in standard English?

Reviewer #1: Yes

Reviewer #2: Yes

5. Review Comments to the Author

Reviewer #1: This is an important study. Some comments

Mixed referencing styles - at times you give authors and date, and at other times you list numerical numbers)

In your introduction you justify orphanages. Perhaps given the enormity of the problems you could give a stronger statement of how this is a negative environment and how despite global guidance orphanages persist (rather than your bland stataement that deinstitutionalized forms of care (perhaps name it as family based care) is not provided!! (Who says it cannot be?) Your next paragraph clearly gives some of the negative issues. Overall the introduction is thorough, clear and well referenced.

The methods are well described and clearly set out. The essential limitation is that the responses are all self report with no validation by the child or even an observer. The possibility of response bias is high and it may well be that the caregivers pick up the social acceptability during the course and respond but there may be a vast gap between attitudes and behaviour. This is a fundamental weakness of this intervention evaluation and needs to be raised and discussed in the discussion.

Discussion. The authors need to insert "reported levels" into all their findings. We do not know if there was reduced levels of maltreatment - only reduced reported levels of maltreatment. It is a worthy finding to see that changed attitudes persisted over time. It would be important to add some reflections on how this may translate into behaviour and perhaps even how a future study may be set up to measure behaviour rather than confined to self reported attitiudes.

It would be crucial that the authors add a paragraph to the discussion on how intolerable such levels of violence are and that trainng is just a small intervention in a crisis, but a higher level longer term child protection lens may be vital to totally reassess this form of care. Vital also to show that despite working in these many institutions over many years there is not a single perspective from a child.

Reviewer #2: This paper presents findings from a cluster randomized controlled trial aiming to replicate and show sustainability of an intervention on interaction competencies with children-for caregivers.

The paper makes an important contribution to the literature on interventions for caregivers and institutional care facilities in Tanzania.

However, there are areas that require clarification.

It seems like the secondary outcomes were significant than the primary ones. Could the authors shed some light on why that was the case?

Structural factors may play a key influence on outcomes within orphanages/institutional care settings and may in some cases interfere with a well designed and implemented intervention. In as much as this was not the focus of this paper, do the authors have a sense of the role of structural factors on their intervention?

Were there any differences in outcomes by institution type? The authors mention that the orphanages/ institutional care facilities were heterogeneous. What exactly do they mean by this? What are some of the differences that were clear within the orphanages and among caregivers that readers need to be aware off in the interpretation of these findings?

The authors need to rework the references throughout the paper. There seems to be more than one referencing style used in the paper. Some references cited in the paper are also missing in the reference list.

6. PLOS authors have the option to publish the peer review history of their article (what does this mean?). If published, this will include your full peer review and any attached files.

**Do you want your identity to be public for this peer review?** For information about this choice, including consent withdrawal, please see our Privacy Policy.

Reviewer #1: No

Reviewer #2: No

---

## [Decision Letter · Decision Letter 1]

21 Feb 2022

PGPH-D-21-00959R1

Sustainability of effects and secondary long-term outcomes: One-year follow-up of a cluster-randomized controlled trial to prevent maltreatment in institutional care

Dear Dr. Hecker,

Thank you for resubmitting your manuscript to PLOS Global Public Health. Unfortunately, the two original reviewers were unavailable to appraise your revision. Subsequently, and additional reviewer was sought. Subsequently, we invite you to submit a revised version of the manuscript that addresses the minor points raised by the third reviewer.

We look forward to receiving your revised manuscript.

Kind regards,

Martin Heine

Academic Editor

Journal Requirements:

1. Please update your Competing Interests statement. If you have no competing interests to declare, please state: “The authors have declared that no competing interests exist.”

Additional Editor Comments (if provided):

Reviewers' comments:

Reviewer's Responses to Questions

**Comments to the Author**

1. If the authors have adequately addressed your comments raised in a previous round of review and you feel that this manuscript is now acceptable for publication, you may indicate that here to bypass the “Comments to the Author” section, enter your conflict of interest statement in the “Confidential to Editor” section, and submit your "Accept" recommendation.

Reviewer #3: (No Response)

2. Does this manuscript meet PLOS Global Public Health’s publication criteria? Is the manuscript technically sound, and do the data support the conclusions? The manuscript must describe methodologically and ethically rigorous research with conclusions that are appropriately drawn based on the data presented.

Reviewer #3: Yes

3. Has the statistical analysis been performed appropriately and rigorously?

Reviewer #3: Yes

4. Have the authors made all data underlying the findings in their manuscript fully available (please refer to the Data Availability Statement at the start of the manuscript PDF file)?

Reviewer #3: Yes

5. Is the manuscript presented in an intelligible fashion and written in standard English?

Reviewer #3: Yes

6. Review Comments to the Author

Reviewer #3: This is a very carefully done study conducted under substantially difficult circumstances. The variability of the institutions and the working conditions is extraordinary and very difficult. Agreed, the study has its limits, mainly that it relies solely on the reports of the trainees. But the authors point out that this is the first step and that this step is necessary before investigating additional steps in improving care. A few specific items:

Page 4, bottom. Ref 9 showed better effects after 9 than 4 months of the intervention.

Page 7, Table 1. Were there any significant differences between the intervention and Waitlist group on these characteristics that might explain any differences between these groups?

Page 17. I wonder if the premature changes in the waitlist group could be because of familiarity with the assessments that tell them what you are looking for and value?

A good job with a very difficult set of circumstances.

7. PLOS authors have the option to publish the peer review history of their article (what does this mean?). If published, this will include your full peer review and any attached files.

**Do you want your identity to be public for this peer review?** For information about this choice, including consent withdrawal, please see our Privacy Policy.

Reviewer #3: No

---

## [Editor Report · Decision Letter 2]

31 Mar 2022

Sustainability of effects and secondary long-term outcomes: One-year follow-up of a cluster-randomized controlled trial to prevent maltreatment in institutional care

PGPH-D-21-00959R2

Dear PhD Hecker,

We are pleased to inform you that your manuscript 'Sustainability of effects and secondary long-term outcomes: One-year follow-up of a cluster-randomized controlled trial to prevent maltreatment in institutional care' has been provisionally accepted for publication in PLOS Global Public Health.

Best regards,

Martin Heine

Academic Editor